# Preliminary efficacy of a community health worker homebased intervention for the control and management of hypertension in Kiambu County, Kenya- a randomized control trial

Grace Wambura Mbuthia[1]*, James Mwangi[2,3], Karani Magutah[4], James Odhiambo Oguta[5], Kenneth Ngure[1], Stephen T. McGarvey[6]

1 College of Health Sciences, Jomo Kenyatta University of Agriculture and Technology, Nairobi, Kenya, 2 School of Public Health and Community Development, Maseno University, Kisumu, Kenya, 3 World Health Organisation Somalia, ATMIS Protected Area, Mogandishu, Somalia, 4 School of Medicine, College of Health Sciences, Moi University Eldoret, Eldoret, Kenya, 5 Division of Population Health, Sheffield Centre for Health and Related Research, School of Medicine and Population Health, University of Sheffield, Sheffield, United Kingdom, 6 Department of Epidemiology, School of Public Health and Department of Anthropology, International Health Institute, Brown University, Providence, RI, United States of America

* grace.mbuthia@jkuat.ac.ke

## Abstract

### Introduction

In Sub Saharan Africa, there is a growing burden of non-communicable diseases, which poses a big challenge to the resource-limited health system in these settings.

### Objective

The aim of this study was to determine the feasibility and preliminary efficacy of a community health workers (CHWs) home-based lifestyle interventions to improve blood pressure (BP) control and body composition among hypertensive patients in low-income populations of Kiambu County, Kenya.

### Methods

This was a randomized controlled trial (RCT) involving 80 patients with uncontrolled high BP (systolic BP (SBP) $\geq$140mmHg and/or diastolic BP (DBP) $\geq$90) randomized to either a CHW homebased intervention or a usual care (control) arm and followed up for 6 months. The intervention involved monthly CHW home-visits for health education and audits on behavioral risk factors that affect BP. An adapted WHO stepwise questionnaire and international physical activity questionnaire was used to collect data on behavioral cardiovascular risk factors. To assess the main outcomes of BP, body mass index (BMI) and waist-height-ratio (WHtR), a survey was conducted at baseline, 3 months, and 6 months. Data regarding univariate, bivariate and multivariate (repeated measurements between and within groups) analysis at 5% level of significance were analyzed using STATA 18. Generalized estimating

**Data Availability Statement:** All relevant data are within the manuscript and its Supporting Information files.

**Funding:** This research was supported by the Consortium for Advanced Research Training in Africa (CARTA). CARTA is jointly led by the African Population and Health Research Center and the University of the Witwatersrand and funded by the Carnegie Corporation of New York (Grant No. G-19-57145), Sida (Grant No:54100113), Uppsala Monitoring Center, Norwegian Agency for Development Cooperation (Norad), and by the Wellcome Trust [reference no. 107768/Z/15/Z] and the UK Foreign, Commonwealth & Development Office, with support from the Developing Excellence in Leadership, Training and Science in Africa (DELTAS Africa) programme. The statements made and views expressed are solely the responsibility of the authors. The funders had no role in study design, data collection and analysis, decision to publish, or preparation of the manuscript.

**Competing interests:** The authors have declared that no competing interests exist.

equations (GEE) for repeated measures were used to estimate changes in BP, BMI and WHtR, and to examine the association between the CHW intervention and BP control.

## Results

The study revealed that 77.5% and 92.5% of the participants in usual care and intervention groups completed the follow-up, respectively. After 6 months of follow-up, there was a reduction in the mean SBP and DBP for both arms, and reductions in BMI and WHtR only in the intervention arm. The adjusted mean reduction in SBP (-8.4 mm Hg; 95% CI, -13.4 to -3.3; P = 0.001) and DBP (-5.2 mm Hg 95% CI, -8.3 to -2.0; P<0.001) were significantly higher in the intervention group compared to the control group. The proportion of participants who achieved the controlled BP target of <140/90 mm Hg was 62.2% and 25.8% for the intervention and usual care arm, respectively. The proportion with controlled BP was significantly higher in the intervention arm compared to the usual care arm after adjusting for baseline covariates (AOR 2.8, 95% CI 1.3–6.0, p = 0.008). There was no significant effect of the intervention on BMI and WHtR.

## Conclusion

A home-based CHW intervention was significantly associated with reduction in BP among hypertensive patients compared to usual care. Future fully powered RCTs to test the effectiveness of such interventions among low-income populations are recommended.

## Trial registration

**Trial registration number:** PACTR202309530525257.

## Background

In Sub Saharan Africa (SSA), there is a growing burden of non-communicable diseases (NCDs), which poses a big challenge to the resource limited health systems in these settings [1–3]. Further, management of cardiovascular diseases (CVDs) in low resource settings is difficult due to limited human and financial resources. There is a need for research to explore innovative, cost effective, and contextually relevant primary care interventions to control BP levels and other cardiovascular risks factors amidst the growing dual burden of infectious and NCDs in SSA [4].

Community health workers (CHWs) also known as community health volunteers or community health promoters are an affordable and sustainable solution for behavioural intervention delivery, and an important linkage between community and health care system[5]. In light of critical shortages in the health workforce in low and medium income countries (LMICs), CHWs, defined as members of a community with minimal formal training on health problems who provide basic health and medical care to their community, are increasingly recognized as an essential part of the health workforce needed to achieve public health goals [6–9].

The CHWs may remove barriers to BP control and medication adherence due to cultural, educational, and language differences between community members and the health care system [10]. A systematic review on the effectiveness of CHWs interventions for management of hypertension in the United States showed significant improvement in BP control particularly

among the poor, urban minority communities [11]. Similarly, community health workers home-based interventions have shown positive impact in the management and control of hypertension in LMICs [12–16]. Another systematic review in LMICs reported positive effects of CHW interventions on improved linkage of patients to care, reduction of BP, improving adherence of patients to medication and the overall reduction in the CVD risk score [17].

Hypertension is the leading risk factor for deaths due to CVDs and as such, the World Health Organization (WHO) targets to have a worldwide 25% reduction in its prevalence by the year 2025 [18]. The 2015 STEPwise survey shows 24% of Kenyans either had elevated BP or were on treatment for hypertension. Only 8% of the hypertensive persons were on treatment, and among them, only 4.6% had controlled BP [19]. Based on literature, modification of lifestyle factors can delay onset of hypertension and can contribute to lowering of BP in treated patients [20,21]. Systematic reviews have shown efficacy of interventions focused on physical activity [22,23] and dietary approaches to stop hypertension [24–26] in lowering BP among adults with or without hypertension in different settings.

One of the ways to address the emerging burden of hypertension could be through home BP monitoring and lifestyle interventions led by CHWs. However, the feasibility and effectiveness of CHWs primary health intervention in the control of hypertension in Kenya is not well explored. Previous research conducted in Western Kenya demonstrated efficacy of CHWs intervention in improving linkage to hypertension health care [12] in the general population. However, there is a lack of uptake of evidence-based community based CHWs interventions for reduction of BP in the Kenyan settings. There is a need for further studies to support primary health interventions geared towards control of hypertension and CVD outcomes among hypertensive patients in the Nairobi metropolitan area where the current study is set. While previous CHWs hypertension interventions in LMICs have focused on screening and health education, the current study proposes to adapt a multicomponent evidence-based lifestyle intervention that incorporates behaviour change communication and practical individualized lifestyle interventions to reduce high BP and other CVD risks among low-income population in Kiambu County. The aim of this study is to test the feasibility and preliminary efficacy of a CHW-led lifestyle homebased intervention for BP reduction among hypertensive patients.

## Methods

### Research design

This was a randomized controlled trial (RCT). Participants from two primary health care facilities were randomized to either the CHW-led homebased intervention or the usual care arm. Randomisation was at facility level. Those assigned to the intervention arm were advised to continue receiving their usual care. The intervention was implemented for 6 months with outcome assessment after 3rd and 6th month.

### Study area

The study was conducted in Kiambu County, Kenya. The county is in the Central Kenyan highlands with an altitude ranging from 1,400 m to 1,800 m above sea level. The county is situated between latitude 0˚75' and 1˚20' south of the equator and longitudes 36˚54' and 36˚85' east. Specifically, the study took place in two level 3 primary health care (PHC) facilities in Juja and Ruiru Sub-counties. The two Sub-counties are predominantly urban and are located in the outskirts of the capital city and within Nairobi metropolitan region.

## Study population

We selected patients with high BP (systolic $\geq$140 mmHg and/or diastolic $\geq$90 mmHg measured on at least 2 separate screening measurements), aged 18 years and above, receiving primary care from the twolevel 3 PHC facilities and available to be followed up for 6 months. Participants commenced on antihypertensive medications were also included. Elderly hypertensive patients aged above 70 years, hypertensive patients already on hypertension follow-up and management elsewhere or wished to relocate from the study area during the study period were excluded.

## Sample size and power calculation

With changes in BP as the primary outcome, we calculated a sample of 52 participants for the 2 arms (26 in each). This would show changes in BP between the intervention and control arms at a significance level of 0.05 for a two-sided test with 80% statistical power of finding a large effect (*Cohen's d* = 0.80). This study was underpowered to find significantly small effect (*Cohen's d* = 0.30) and moderate effect (*Cohen's d* = 0.50). However, we were able to calculate an effect size and show the direction of its impact. After adjusting for 20% loss to follow-up, the total sample was 62 participants.

## Sampling and participant's recruitment

The participants were consented at Ruiru and Juja level 3 primary health facilities following referral from a community-based door-to-door BP screening in the study area (Fig 1). The two primary health facilities were in 2 different sub-counties where home blood pressure screening took place and participants were referred to the appropriate facility in their respective Sub-counties The recruitment took place from October to November 2022. Participants with high BP (systolic $\geq$140 mmHg and/or diastolic $\geq$90 mmHg) were given a referral note to either Juja or Ruiru primary health facility. After clinician's assessment and commencement of management, participants with high BP willing to participate were recruited into the study. For the purposes of the pilot RCT only 40 participants were recruited in each facility. Simple random sampling (two folded papers bearing the name of the two facilities were tossed and one picked at random became the intervention facility) was used to allocate the two PHC facilities (and the patients receiving care there) to either the control usual care or CHW home-based intervention arms.

## Study procedure

The research assistants completed a baseline assessment for the eligible participants. **Intervention:** The community health workers were trained to coach patients on lifestyle modification, home BP-monitoring, and medication adherence during a 2-day interactive training session followed by onsite field return demonstrations. The CHW were taught culturally appropriate behavioral change related to hypertension based on the Centers for Disease Control(CDC) CHW training manual for preventing heart disease and stroke [27]. Before the actual intervention, several mock intervention sessions were done and critiqued to ensure CHW ability to deliver intervention content consistently. Additionally, the research assistants (qualified nurse) made monthly supervisory visits and sat in the intervention sessions to ensure that home visits and intervention were delivered effectively. Those in the intervention arm received a home based community health worker–led intervention (health coaching, home BP monitoring, BP audit and feedback) implemented over a period of 6 months. This entailed two home visits by the trained CHWs during the first month after randomisation, followed by

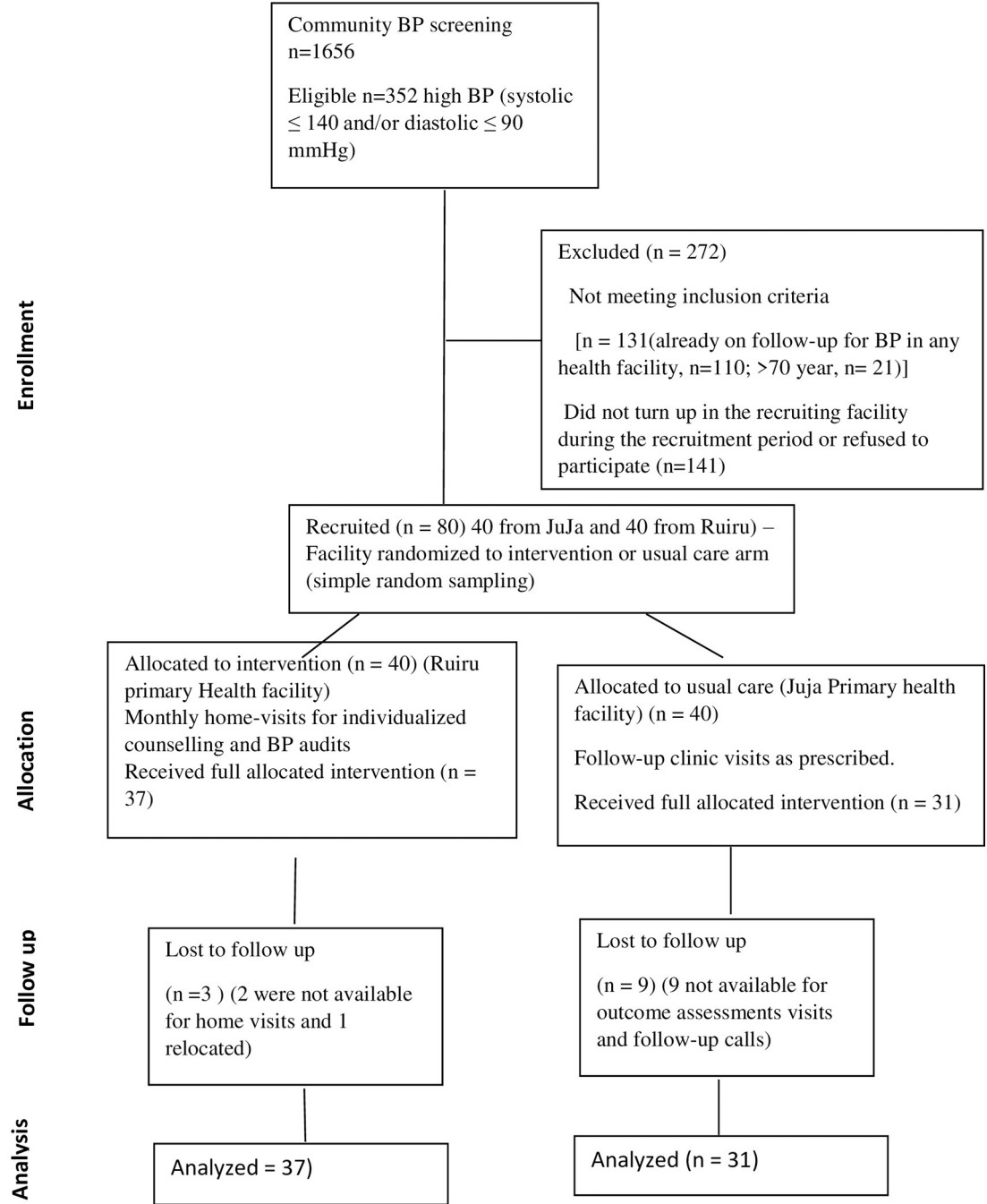

**Fig 1. Consolidated Standards of reporting trials describing recruitment and completion of study procedures.**

monthly follow-up visits for a period of 6 months. The initial visit was a 90-minute home visit to discuss general knowledge about hypertension and offer tailored counselling on lifestyle modification as well as set PA targets. Subsequent monthly visit was focused on social support, goal setting, BP, and weight monitoring. Participants in the intervention arm were also encouraged to adhere to the usual care by attending the prescribed clinic visits for hypertension at the primary health care facility.

The CHWs were trained to adapt the counselling based on individual participant needs by emphasizing on actions to be taken based on hypertension risk factors identified. Subsequent monthly visits were focused on social support, goal setting, BP, and weight monitoring. For motivation, the CHW were paid the Ministry of Health recommended monthly stipend for CHW of approximately 60 USD per month.

The participants in the control arm received the usual care, which included monthly clinic visits to the health facility and medical management as prescribed by the clinicians. Participants in this arm did not have home visits by CHW.

To assess the outcome of the intervention, home-based surveys were conducted by two trained research assistants (who were not part of nor had relationship with the CHWs team) to evaluate the participants at baseline, 3 months, and 6 months. The research assistants were blinded to the randomization status of the participants. The home visits for evaluation were scheduled during the first half of the day to minimize the effect of diurnal variations in BP.

## Data collection

An adapted WHO stepwise questionnaire was used to collect information on demographic characteristics and health behaviours (smoking, alcohol drinking, diet and physical activity (PA)) of participants at baseline and follow-up visits. The international PA questionnaire (IPAQ) [28] was used to collect data on PA. Three BP measurements, body weight, height and waist circumference were obtained at each data collection visit by outcome assessors masked to intervention assignment. Blood pressure was measured using a digital (Omron M1) BP machine according to American Heart Association Guidelines. The guidelines include resting seated for 5 minutes prior to monitoring, abstaining from smoking, drinking, and exercise 30 minutes prior, and recording three measurements each time the monitor is used and an average obtained [29]. Weight was measured using a bathroom scale (CAMRY Mechanical scale, BR9012, Shanghai, China) with the subject in light clothing and without shoes.

The primary outcomes were the differences in systolic and diastolic BP (SBP and DBP respectively) changes from baseline to the 6 month follow-up survey while secondary outcomes included the proportion of patients with controlled hypertension (BP <140/90 mm Hg) according to the Kenyan national guidelines [30], body composition–body mass index (BMI) [31], and waist height ratio (WHtR) [32]. The outcome measures were assessed at baseline, 3 months, and 6 months of follow-up. Physically active was defined as having attained 150 minutes of moderate PA or 75 minutes of vigorous PA each week.

## Statistical analysis

The *Stata* 18 software (Stata Corporation, College Station, Texas, USA) was used for data (S3 File) entry and analysis. Descriptive statistics were used for participants' characteristics. Categorical data were analyzed and reported in frequency and percentages. Independent samples t-test was used to determine whether baseline characteristics (continuous variables) differed significantly between the control and the usual care arms at baseline while differences in the categorical variables between the arms were analyzed using a test for 2-sample differences in proportions. The mean changes in BP and BMI were first estimated with using paired t-test for the difference between end line and baseline measurements.To account for repeated BP and BMI measurements, we used the generalized estimating equations (GEE) model for repeated measures using gaussian identity link function while adjusting for baseline characteristics; antihypertensive treatment and demographic characteristics. The GEE model was also used to examine the association between the CHW intervention with BP control (BP less than 140/90mmHg) and normal BMI(BMI 18.5–24.9kg/m2), specifying logit link function and

reporting odds ratios, after controlling for baseline BP, baseline BMI, baseline WHtR, age, sex, and use of antihypertensive medication. We selected the GEE model because it is a population average model that does not rely on probability distribution assumptions [33]. However, we used the mixed effects models as sensitivity analyses for our estimates of the intervention effect (S1 Table). Statistical significance was considered at P < 0.05.

### Ethical considerations

The study proposal (S2 File) was approved by Jomo Kenyatta University of Agriculture and Technology (JKUAT) Institutional scientific and ethical review committee (approval number JKU/IERC/02316/0652) [S4 File]. Similarly, research permit was obtained from the National Commission of Science, Technology, and Innovation before commencement of the study (license number NACOSTI/P/22/19977). Confidentiality and anonymity of patients was guaranteed by excluding unique identifiers from the data collected from participants. Participation in the study was on voluntary basis and written informed consent was obtained from the participants before data collection.

This study is registered in the Pan African Clinical Trial Registry database, registration number PACTR202309530525257.

## Results

A total of 80 participants (40 in each arm) were recruited in the study. Of these, 31(77.5%) of the participants in usual care and 37(92.5%) in the intervention arm completed the follow-up (Fig 1). The mean age of the participants was 46.8±11.1 years and ranged from 25–68 years. At baseline, the mean SBP, DBP and BMI were 155.7 mmHg, 97.1 mmHg and 29.7 kg/m$^2$, and 151.7 mmHg, 95.7 mmHg, 29.3 kg/m$^2$ for the usual care and intervention arm respectively. A majority of the participants were physically active in both arms with 81% and 90% in the intervention and usual care arm respectively, reporting having achieved at least 75 minutes of vigorous intensity or 150 minutes of moderate intensity activity per week. Only 4(6%) of the participants smoked cigarettes and a quarter 17(25%) consumed alcohol. At baseline, approximately half of the respondents, 56.8% in the intervention and 45.2% in the usual care arms, were on antihypertensive treatment. There were no statistically significant differences in the baseline characteristics between the two arms (Table 1).

### Implementation and adherence to the intervention

Implementation and adherence to the usual care was assessed through the attendance of health facility follow-up visits for both arms. A third, 25(62.5%) of the participants in the usual care attended all the prescribed six health facility follow-up visits while six participants attended five visits and nine participants attended less than four clinic visits. On the other hand, 28 (70%) of the participants in the intervention arm completed all the six prescribed health facility follow-up visits while seven participants attended five visits and five participants attended less than 4 visits.

In addition to the usual care, the intervention arm received monthly home visits by the CHW for a period of 6 months. A majority (92.5%) received all the planned home visits while the rest only completed 33% of the home visits and were not available for outcome assessment thus were lost to follow-up. Using a health education checklist (S1 File) the initial visit was focused on general knowledge about hypertension and tailored counselling on lifestyle modifications on PA and diet (reduction of salt intake, increase fruits and vegetables and reduction of deep fried foods) as well as goal setting. Subsequent monthly visits were focused on BP and weight audits, review on achievement of at least 150 minutes of moderate PA and counselling

**Table 1. Baseline characteristics of the participants.**

| Variable | Usual Care arm (N = 31) Mean (SD) or n (%) | Lifestyle Intervention arm (N = 37) Mean (SD) or n(%) | Absolute difference (95% CI) | P- Value |
|---|---|---|---|---|
| Age in years | 46.3(9.7) | 47.3(14.4) | -0.55(-6.04–4.93) | -0.841[†] |
| SBP(mmHg) | 155.7(18.3) | 151.7(10.4) | 4.03(-3.04–11.1) | 0.258[†] |
| DBP(mmHg) | 97.1(11.8) | 95.7(11.2) | 1.42(-4.17–7.01) | 0.613[†] |
| BMI(kg/m$^2$) | 29.7(5.5) | 29.3(6.1) | 0.50(-2.3–3.32) | 0.722[†] |
| WHtR | 0.59(0.12) | 0.61(0.10) | -0.01(-0.06–0.04) | 0.682[†] |
| **Gender** | | | | |
| Male | 5(16.1) | 5(13.5) | -0.02(-0.19–0.14) | 0.761[*] |
| Female | 26(83.9) | 32(86.5) | | |
| **Physically active (achieved 75 minutes of vigorous intensity or 150 minutes of moderate intensity activity** | | | | |
| No | 3(9.7) | 7(18.9) | 0.09(-0.07–0.25) | 0.284[*] |
| Yes | 28(90.3) | 30(81.1) | | |
| **Smoking** | | | | |
| No | 28(90.3) | 36(97.3) | 0.06(-0.04–0.19) | 0.223[*] |
| Yes | 3(9.7) | 1(2.7) | | |
| **Alcohol Use** | | | | |
| No | 22(70.9) | 29(78.4) | | |
| Yes | 9(29) | 8(21.6) | 0.07(-0.13–0.28) | 0.482[*] |
| **On anti-hypertensive** | | | | |
| No | 17(54.8) | 16(43.2) | -0.11(-0.35–0.12) | 0.341[*] |
| Yes | 14(45.2) | 21(56.8) | | |
| **Consumption of deep fried foods** | | | | |
| Rarely | 20(64.5) | 21(56.8) | | |
| Frequently(more than twice a week) | 11(35.5) | 16(43.2) | -0.08(-0.31–0.15) | 0.514 |

**SBP**- Systolic blood pressure **DBP**- Diastolic blood pressure **BMI**- Body mass index **WHtR**- Waist height ratio.

[†] P value for differences in continuous variables was generated using independent samples t-test.

[*]P values for categorical variables was generated using a test for 2-sample differences in proportions.

on diet and treatment adherence (for those on anti-hypertensives). Thirty-one (77.5%) of the participants in the usual care arm were available for the 3 home visits focused on outcome assessments while 5 and 4 participants were available for one and two visits, respectively.

## Blood pressure and body composition outcomes

Table 2 presents the mean measurements of the sample at different time points. After 6 months of follow-up, there was a reduction in the mean SBP, DBP in all the arms and a reduction in BMI and WHtR only among the intervention arm (Table 2).

The mean SBP in the intervention arm reduced by 19 mm Hg (95% CI -21.04 to -16.9), while in the usual care arm it had reduced by 7.9 mm Hg (95% CI -11.8 to -4.0). Similarly, the mean unadjusted DBP fell by 9.9 mm Hg (95% CI -13.1 to -6.6) in the intervention arm and 3.8 mmHg (95% CI -8.5 to -0.9) in the usual care arm (Fig 2). The mean BMI increased by 0.40 kg/m$^2$ in the usual care arm whereas in the intervention arm it reduced by –0.90 kg/m$^2$. Similarly, the mean WHtR in the usual care arm increased by 0.003 while that of the intervention arm reduced by -0.02 as shown in table two.

**Table 2. Mean blood pressure and body composition measures over 6 months.**

| Variable | Arm | Baseline Mean (SD) | Month 3 Mean (SD) | Month 6 Mean (SD) |
|---|---|---|---|---|
| SBP (mmHg) | Usual Care | 155.7(18.3 | 155.3(13.6) | 147.8(18.1) |
| | Intervention | 151.7(10.4) | 140.8(17.1) | 132.7(9.5) |
| DBP (mmHg) | Usual Care | 97.1(11.8) | 98.9(7.8) | 93.3(10.1) |
| | Intervention | 95.7(11.2) | 90.8(11.7) | 85.7(7.7) |
| BMI (kg/m²) | Usual Care | 29.7(5.5) | 29.4(4.8) | 30.1(4.9) |
| | Intervention | 29.3(6.1) | 28.9(5.7) | 28.4(5.4) |
| WHtR | Usual Care | 0.60(0.1) | 0.62(0.1) | 0.61(0.1) |
| | Intervention | 0.61(0.1) | 0.61(0.1) | 0.59(0.1) |

**SD**- Standard deviation **SBP**- Systolic blood pressure **DBP**- Diastolic blood pressure **BMI**- Body mass index WHtR- Waist height ratio.

## Between group differences in mean changes

Table 3 presents the difference in the net changes in BP, BMI and WHtR between the intervention groups. The adjusted mean reduction in SBP was 8.4 mm Hg greater in the intervention group than in the control group (95% CI, -13.4 to -3.3; P = 0.001) while the adjusted mean reduction in DBP in the intervention arm compared to the control {5.2 mm Hg (95% CI, -8.3 to -2.0; P<0.001)} (Table 3). However, the observed adjusted net mean difference in BMI

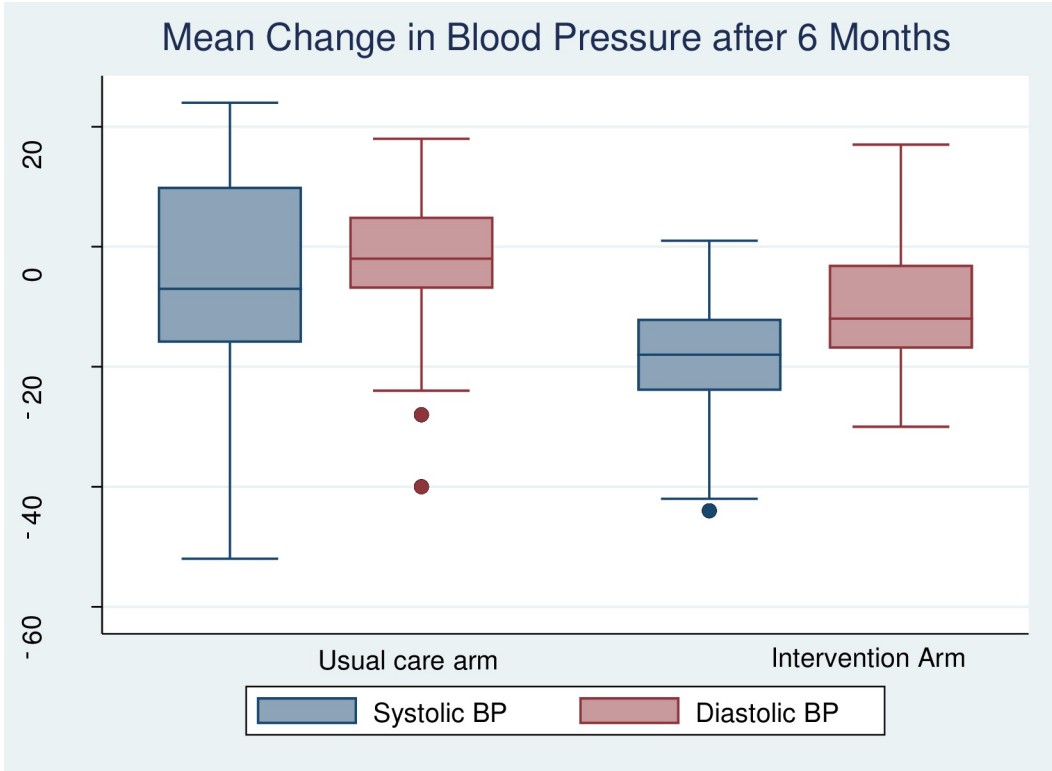

**Fig 2. Mean change in blood pressure after 6 months of follow-up.**

**Table 3.  Effect of the intervention on mean changes in blood pressure and body composition measures after 6 months of follow-up.**

| Variable | Arm | Unadjusted mean change within each arm (95% CI) | Unadjusted net mean change between the arms (95% CI) | P Value | Adjusted net mean change (95% CI) | P value |
|---|---|---|---|---|---|---|
| SBP (mmHg) | Usual Care | -7.9(-11.8 to—4.0) | -11.2 (-16.9 to -5.6) | <0.001 | -8.4(-13.4 to -3.3) | 0.001 |
| | Intervention | -19(21.04 to -16.9) | | | | |
| DBP (mmHg) | Usual Care | -3.8 (-8.5to -0.9) | -5.7 (-9.5 to -1.8) | 0.004 | -5.2 (-8.3 to -2.0) | 0.001 |
| | Intervention | -9.9(-13.1 to -6.6) | | | | |
| BMI (kg/m²) | Usual Care | 0.40(-0.002–0.81) | -1.0 (-3.5 to 1.6) | 0.453 | -1.5 (-3.7 to 0.6) | 0.158 |
| | Intervention | -0.90(-1.3 to -0.49) | | | | |
| WHtR | Usual Care | 0.003(-0.01 to 0.01) | -0.006 (-0.04 to 0.04) | 0.792 | 0.0 (-0.03 to 0.03) | 0.999 |
| | Intervention | -0.02(-0.03 to -0.01) | | | | |

SBP- Systolic blood pressure DBP- Diastolic blood pressure BMI- Body mass index WHtR- Waist height ratio.

For change values, negative number demonstrates improvement. Adjusted analysis were conducted using GEE (specifying gaussian family and identity link function) adjusted for age, gender, BP at baseline, use of antihypertensive medications, BMI and WHtR at baseline.

(-1.5; 95% CI, -3.7 to 0.6; P = 0.158) and WHtR (0.0; 95% CI, -0.03 to 0.03; P = 0.999) between the two groups were not statistically significant.

The proportions of participants who achieved the controlled BP target of <140/90 mm Hg were 62.2% (23) and 25.8% (8) for the intervention and usual care arms, respectively. The odds of attaining blood pressure control in the intervention arm were significantly higher compared to the usual care arm after adjusting for baseline BMI, baseline WHtR, age, gender and being on antihypertensive drugs (AOR 2.8, 95% CI 1.3–6.0, p = 0.008) (Table 4). However, we did not observe a significant difference in the odds of normal BMI between the treatment arms (AOR 2.5, 95% CI 0.6–10.4, p = 0.181) (Table 4).

### Adverse outcome

There were no adverse outcomes (deaths) reported during the study.

## Discussion

Our study provides evidence that a homebased lifestyle and BP monitoring intervention, delivered by CHWs, was more effective in reducing SBP and DBP and in improving control of hypertension in low-income urban population, compared to usual care. The completion rates of the follow-up were favorable for both the usual care (77%) and intervention arms (92%), indicating the feasibility of implementing such interventions among an urban population. The better completion rates for the intervention group may be attributed to the interest among the participants cultivated through the regular CHW home visits and audits of the progress that was not the case in the usual care arm.

**Table 4.  Difference in odds of attaining BP control and normal BMI between treatment arms.**

| Variable | Arm | Unadjusted net mean change between the arms (95% CI) | P Value | Adjusted net mean change (95% CI) | P value |
|---|---|---|---|---|---|
| BP Controlled (Yes/No) | Usual Care (reference) | 3.2 (1.5–6.9) | 0.004 | 2.8 (1.3–5.9) | 0.008 |
| | Intervention grp | | | | |
| Normal BMI (Yes/No) | Usual Care | 1.3 (0.4–3.8) | 0.634 | 2.6 (0.6–10.4) | 0.181 |
| | Intervention grp | | | | |
| | Intervention grp | | | | |

After 6 months of follow-up, the study showed a significantly greater reduction in the mean adjusted SBP and DBP among the intervention than the usual care arm. Our findings are consistent with other studies[14,16,33] on the effect of lay health worker intervention programs for control of BP in LMICs. Our findings compare to those in a study done in Nepal with CHWs home-monitoring and education, which found 4.9mmHg and 2.6 mmHg greater decrease in SBP and DBP, respectively, for the intervention arm compared to usual care arm after one year of follow-up [14]. Similarly, a study done in Argentina among low-income population showed that participants with uncontrolled hypertension who participated in a community health worker–led multicomponent intervention had 6.6 and 5.4 mmHg greater decrease in SBP and DBP, respectively, than did patients who received usual care after 18 months of follow-up [16].The personalised health education and BP audits delivered by home visits in the current study might have encouraged participants to adopt healthy lifestyles and adhere to physicians instructions leading to controlled BP. Additionally, the CHW intervention comprised components that are evidence-based for lowering BP. These included regular home BP audits [34], encouraging lifestyle changes of increased PA and losing weight [35] as well as encouraging adherence to antihypertensive drugs for those on treatment [36].

The current study showed a reduction in BMI and WHtR in the intervention arm, compared to slight increase in the usual care arm. However, we did not observe a significant difference in the mean change BMI and WHtR between the treatment arms. Our findings are consistent with those of Gamage et al (2020) and He et al (2017) that showed a CHW-led intervention for control of hypertension had no effect on BMI and WHR [16,33]. Our findings could be attributed to the short period of follow-up of six, which may not have resulted in significant changes in body weight to affect the body composition measures.

This feasibility study had some limitations. First, the duration of the intervention was limited to 6 months, and therefore, it is difficult to determine the extent to which changes in BP and body composition could be sustained. Additionally, we cannot predict the effect of the intervention after 6 months. Secondly, as a feasibility study with limited resources, we recruited a small sample size. However, the findings are crucial in informing the design of a fully powered RCT. Thirdly, the study may have had measurement bias for anthropometric measurements, but this was minimized through training of the enumerators on standard procedures and their support supervision. We also did not account for any potential differences in the quantity or quality of usual care received by participants in both arms at their respective health facilities. Fourthly, there was a difference in the number of participant on anti-hypertensive treatment at baseline that could have been a source of bias, however this difference was not statistically significant and the use of anti-hypertensive was adjusted for in the multivariate model. Finally, given that this was a feasibility study, we did not monitor adherence to antihypertensive treatment and longitudinal changes in health behaviours that could be attributed to the changes in blood pressure.

The strengths of the study are that, to the best of our knowledge, this is the first study investigating the feasibility and effectiveness of CHW home-delivered interventions for the reduction of BP among a poor urban population in Kenya. Secondly, the baseline and follow-up outcome assessment surveys were done by independent outcome assessors and the key BP outcome was measured using an automated device to minimize the risk of assessor's bias. Thirdly, the study has been reported following the Consolidated Standards of Reporting Trials (CONSORT) guidelines (S5 File).

## Conclusion

A CHW home based lifestyle intervention was effective in reducing BP and had a higher proportion of controlled hypertension compared to the usual care. These findings add to the body

of knowledge on task-shifting interventions for treating high BP in LMICs. In particular, the study showed that utilization of CHWs from the already existing health-care systems of PHC in Kenya for management of hypertension is feasible in urban low-income settings of Kiambu County. However, the current findings are only based on a small sample derived from a Kenyan population and therefore future studies with much larger sample sizes and collected at other geographical areas to test the effectiveness of such interventions are recommended.

## Supporting information

**S1 Table. A mixed effects model (using random effects).**
(DOCX)

**S1 File. Education checklist.**
(DOCX)

**S2 File. Trial proposal.**
(DOCX)

**S3 File. Data set.**
(XLSX)

**S4 File. Ethical approval.**
(PDF)

**S5 File. CONSORT-2010 checklist.**
(DOC)

## Acknowledgments

Our special thanks go to the participants in this study for their invaluable contributions to this study. We thank the community health workers who conducted the intervention. We also thank the Sub-county management teams for their assistance during the data collection

## Author Contributions

**Conceptualization:** Grace Wambura Mbuthia, James Mwangi, Karani Magutah, Kenneth Ngure, Stephen T. McGarvey.

**Data curation:** Grace Wambura Mbuthia.

**Formal analysis:** Grace Wambura Mbuthia, Karani Magutah, James Odhiambo Oguta, Stephen T. McGarvey.

**Funding acquisition:** Grace Wambura Mbuthia.

**Investigation:** Grace Wambura Mbuthia.

**Methodology:** Grace Wambura Mbuthia, James Mwangi, Karani Magutah, Stephen T. McGarvey.

**Project administration:** Grace Wambura Mbuthia, Kenneth Ngure.

**Software:** James Odhiambo Oguta.

**Supervision:** Kenneth Ngure, Stephen T. McGarvey.

**Validation:** Grace Wambura Mbuthia, James Odhiambo Oguta.

**Visualization:** Grace Wambura Mbuthia, James Mwangi, James Odhiambo Oguta, Stephen T. McGarvey.

**Writing – original draft:** Grace Wambura Mbuthia.

**Writing – review & editing:** Grace Wambura Mbuthia, James Mwangi, Karani Magutah, James Odhiambo Oguta, Kenneth Ngure, Stephen T. McGarvey.

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
