## [Decision Letter · Decision Letter 0]

5 Dec 2023

PONE-D-23-33530Preliminary efficacy of a community health worker homebased intervention for the control and management of hypertension in Kiambu County, Kenya- A randomized control trialPLOS ONE

Dear Dr. Mbuthia,

Thank you for submitting your manuscript to PLOS ONE. After careful consideration, we feel that it has merit but does not fully meet PLOS ONE’s publication criteria as it currently stands. Therefore, we invite you to submit a revised version of the manuscript that addresses the points raised during the review process.

We look forward to receiving your revised manuscript.

Kind regards,

Patricia Khashayar

Academic Editor

PLOS ONE

Journal Requirements:

2. We note that you have selected “Clinical Trial” as your article type. PLOS ONE requires that all clinical trials are registered in an appropriate registry (the WHO list of approved registries is at      https://www.who.int/clinical-trials-registry-platform/network/primary-registries"" https://www.who.int/clinical-trials-registry-platform/network/primary-registries" https://www.who.int/clinical-trials-registry-platform/network/primary-registries and more information on trial registration is at http://www.icmje.org/about-icmje/faqs/clinical-trials-registration/).

Please state the name of the registry and the registration number (e.g. ISRCTN or ClinicalTrials.gov) in the submission data and on the title page of your manuscript.

a) Please provide the complete date range for participant recruitment and follow-up in the methods section of your manuscript.

b) If you have not yet registered your trial in an appropriate registry, we now require you to do so and will need confirmation of the trial registry number before we can pass your paper to the next stage of review. Please include in the Methods section of your paper your reasons for not registering this study before enrolment of participants started. Please confirm that all related trials are registered by stating: “The authors confirm that all ongoing and related trials for this drug/intervention are registered”.

Please see http://journals.plos.org/plosone/s/submission-guidelines#loc-clinical-trials for our policies on clinical trials.

4. Thank you for stating the following financial disclosure: "This research was supported by the Consortium for Advanced Research Training in Africa (CARTA). CARTA is jointly led by the African Population and Health Research Center and the University of the Witwatersrand and funded by the Carnegie Corporation of New York (Grant No. G-19-57145), Sida (Grant No:54100113), Uppsala Monitoring Center, Norwegian Agency for Development Cooperation (Norad), and by the Wellcome Trust [reference no. 107768/Z/15/Z] and the UK Foreign, Commonwealth & Development Office, with support from the Developing Excellence in Leadership, Training and Science in Africa (DELTAS Africa) programme. The statements made and views expressed are solely the responsibility of the authors."

Reviewers' comments:

Reviewer's Responses to Questions

**Comments to the Author**

1. Is the manuscript technically sound, and do the data support the conclusions?

Reviewer #1: Partly

Reviewer #2: Yes

Reviewer #3: Partly

2. Has the statistical analysis been performed appropriately and rigorously? 

Reviewer #1: No

Reviewer #2: No

Reviewer #3: Yes

3. Have the authors made all data underlying the findings in their manuscript fully available?

Reviewer #1: Yes

Reviewer #2: Yes

Reviewer #3: Yes

4. Is the manuscript presented in an intelligible fashion and written in standard English?

Reviewer #1: Yes

Reviewer #2: Yes

Reviewer #3: No

5. Review Comments to the Author

Reviewer #1: See attached file for cleaner formatting of reviewer comments.

This study is a small pilot randomized trial of a community health worker intervention to deliver lifestyle counseling to people with uncontrolled hypertension identified on community screening in Kiambu County, Kenya. The intervention consisted of lifestyle counseling which focused on dietary changes and exercise recommendations and was delivered via monthly visits by a community health worker. All hypertension care was provided at health facilities and was delivered in the same way for intervention and control participants. The study is well written and makes an important contribution. I do think that more detail is needed on the CHWs themselves and potential mechanisms for effects seen. I also think the choice of cluster randomization is a major limitation and is not adequately accounted for in the analysis or in the discussed limitations of the study. In randomizing only two clinic sites, the sample size is effectively n=2 which limits inferences that can be made

Major

1. The manuscript would benefit from additional detail on the CHW and the behavior change intervention. It would be helpful to include information on the following:

a. What training did CHWs receive (including the content, duration, and frequency of training)?

b. How were the CHWs compensated and how did this compare to current standard?

c. How were CHWs supervised? Was there any quality assessment to assess fidelity of the content being delivered?

d. This is framed as a behavior change intervention. Was there a behavior change framework used to guide CHW training material development?

e. How was the lifestyle counseling material developed and/or adapted for contextual relevance? How did CHWs adapt advice based on individual participant needs?

2. It appears that randomization was done at the facility level, rather than the patient level, which would yield a sample size of n=2 (methods, line 151-153). This is a major limitation and should be more clearly written in the methods, along with justification for why cluster randomization was chosen, as opposed to individual randomization for this small pilot study. Additionally, the authors should explain how clustering is addressed in the sample size calculations and the analysis. Cluster level randomization should also be mentioned in the limitations as it introduces several potential confounders of study findings. For example, if clinicians at the intervention facility were more assertive about titrating antihypertensive medications, this could have explained observed differences in blood pressure.

3. Line 150 indicates that participants were allocated to either Juja or Ruiru health facilities. Can the authors elaborate on how the assigned facility was chosen for each participant? Presumably this was by proximity to the participant’s home, but the fact that there were identical numbers of patients in each clinic makes me wonder if clinics were assigned in some way. This needs additional clarity.

4. The results on blood pressure reduction are impressive given that intervention was lifestyle counseling only and there were high baseline levels of physical activity. Presumably the mechanisms of change were driven by dietary changes, medication adherence, or both. Is there any information available regarding initiation of antihypertensive medications? It seems this was assessed at least by self-report in the research team surveys and would be helpful to know if the intervention arm received medications more frequently than control participants.

5. In addition to medications, can the authors present data on any other factors that may have changed as a result of the intervention? The current manuscript provides details on clinical outcomes (SBP/DBP, BMI), but it is unclear what mechanisms these changes were achieved through. Presentation of changes in diet and exercise as assessed in the surveys would be helpful. I recognize that formal mediation analysis is not possible in this small pilot study, but observations of changes in diet or exercise could be hypothesis generating for a larger trial.

6. The authors note differential follow-up by trial arm in the research surveys that were conducted independent of clinical care engagement. This should be mentioned in the limitations section. Differences seen here could be a function of the intervention keeping people more engaged over time (thus would bias results towards the null and true intervention effects may be even greater than the authors observed), or could be due to underlying differences in the populations given the cluster randomization.

Minor:

1. In the abstract and 1st introduction sentence, I suggest using the term “resource limited health systems” instead of “weak health systems” as this centeres the description on the primary challenge of resources rather than critique of the health workers/systems themselves.

2. In the methods, I would suggest describing the context of Kiambu County and the included health facilities to facilitate reader understanding of applicability of their setting. Simple description of the context as urban, peri-urban, etc , along with differences between the communities surrounding the two clinics would be helpful.

3. Figure 1: there is an error in the first box – should read “Eligible n=352 high BP (Systolic ≥140 and/or diastolic ≥90 mmHg)”

4. Figure 1: The exclusion criteria in Figure 1 indicate that participants were excluded if they are already engaged in hypertension care at ‘any health facility’ but the text indicates that participants were excluded if in care at a different health facility (other than Ruiru or Juja). Please clarify the criteria and harmonize language between the text and Figure 1.

5. Table 1: do not add up to column total for some characteristics in the usual care arm (gender, smoking) – were there missing data on these characteristics?

Reviewer #2: This manuscript presents data analysis from a randomized control trial (RCT) to compare the effectiveness of the CHW homebased intervention (compared to usual care), for control and management of hypertension, in a Kenyan population. The topic is of importance, the study was registered as a RCT (with a valid number), and was approved by the respective IRB/Ethics Committee. While the study objectives sound interesting, is important, and on target, some shortcomings were observed, in regards to abiding by the CONSORT guidelines for conducting and reporting results of high-quality randomized controlled trials (RCTs). Some other (statistical) comments were also provided.

1. Abstract:

It's better for the readers, if the Abstract was written in sections, such as Objectives, Methods, Results and Conclusion.

2. Methods:

Methods reporting need some work. An orderly manner is suggested, following CONSORT guidelines, without repeating information, such as Trial Design, Participant Eligibility Criteria and settings, Interventions, Outcomes, sample size/power considerations, Interim analysis and stopping rules, Randomization (details on random number generation, allocation concealment, implementation), Blinding issues, etc, should be mentioned. The authors are advised to create separate subsections for each of the possible topics (whichever necessary), and that way produce a very clear writeup. They are advised to write it carefully, following nice examples in the manuscript below:

https://www.sciencedirect.com/science/article/pii/S0889540619300010

Specific comments:

(a) For instance, the randomization and allocation concealment should be made very clear (they are NOT the same thing); the trial staff recruiting patients should NOT have the randomization list. Randomization should be prepared by the trial statistician, and he/she would not participate in the recruiting.

(b) The manuscript does not provide details on the randomization procedure; it's stated that a simple random was done. More details on that was done would be helpful, like how was the random sequence generated. Furthermore,

any reasoning, why a block randomization was not used, which is often recommended to ensure a balance in sample size across groups?

https://www.ncbi.nlm.nih.gov/pmc/articles/PMC2267325/

(c) Statistical Analysis:

(c1) The "Data Analysis" section should be named "Statistical Analysis".

(c2) While assessing baseline characteristics, t-test was used which is heavily based on normality assumptions. Under violations, alternative nonparametric tests, such as Wilcoxon rank sum tests, should be mentioned.

(c3) The GEE technique was used to handle repeated measures; any reasoning, why a mixed model (using random effects) was not used?

(c4) GEE with logit link was used; it's better to clearly write how the binary (response) variable was coded.

3. Results & Conclusions:

(a) The authors should check that any statement of significance should be followed by a p-value in the entire Results section. Otherwise, the Results section look OK; it's pretty straightforward.

(b) Conclusions should state that the current findings are ONLY based on the random samples derived from an Iranian population, and should allude to future studies with much larger sample sizes and collected at other geographical areas to confirm the effectiveness of the homebased intervention.

Reviewer #3: I reviewed the manuscript entitled “Preliminary efficacy of a community health worker homebased intervention for the control and management of hypertension in Kiambu County, Kenya- A randomized control trial”. The manuscript is in the scope of the journal, but certain shortcomings should be addressed before the article could be published:

• The manuscript should be edited for language, as it contains several grammar mistakes.

• The introduction section is too long. Please rewrite it.

• The inclusion and exclusion criteria should be clarified.

• The study protocol needs some clarification. The questionnaire filled out at the beginning of the study might have increased the awareness in the control group, have the authors noticed any changed in the lifestyle and behaviour if the control group.

• As for the people in the intervention group who did not adhere to the plan, was there any difference in the BP changes over time with those following the recommendations?

• There is no information on the changes in the behaviour and lifestyle of the intervention group. It is recommended to add those information and also assess how they have affected the final outcome.

6. PLOS authors have the option to publish the peer review history of their article (what does this mean?). If published, this will include your full peer review and any attached files.

Reviewer #1: No

Reviewer #2: No

Reviewer #3: **Yes: **Pouria Khashayar

---

## [Author Response · Author response to Decision Letter 0]

26 Jan 2024

Dear Editor/Reviewers,

We wish to thank you for the comments provided during the review , we have done our best to all of them as outlined in our rebuttal letter.

Concerning funding information and financial disclosure being different, we wish to note that this work was funded by Consortium for Advanced Training in Africa (CARTA) which is funded by different funders outlined in the disclosure. However when filling the funders in the last section it was not possible to choose CARTA as a funder since it is not one of the listed funders. We did not list the grants numbers given in this disclosure since we did not receive funds directly from these funding agencies but we received from CARTA and acknowledge the funders as instructed by CARTA.

 Looking forward for your consideration.

Thank you.

---

## [Decision Letter · Decision Letter 1]

13 Feb 2024

Preliminary efficacy of a community health worker homebased intervention for the control and management of hypertension in Kiambu County, Kenya- A randomized control trial

PONE-D-23-33530R1

Dear Dr. Mbuthia,

We’re pleased to inform you that your manuscript has been judged scientifically suitable for publication and will be formally accepted for publication once it meets all outstanding technical requirements.

Kind regards,

Patricia Khashayar

Academic Editor

PLOS ONE

Additional Editor Comments (optional):

Reviewers' comments:

Reviewer's Responses to Questions

**Comments to the Author**

1. If the authors have adequately addressed your comments raised in a previous round of review and you feel that this manuscript is now acceptable for publication, you may indicate that here to bypass the “Comments to the Author” section, enter your conflict of interest statement in the “Confidential to Editor” section, and submit your "Accept" recommendation.

Reviewer #1: All comments have been addressed

Reviewer #2: All comments have been addressed

2. Is the manuscript technically sound, and do the data support the conclusions?

Reviewer #1: Yes

Reviewer #2: (No Response)

3. Has the statistical analysis been performed appropriately and rigorously? 

Reviewer #1: I Don't Know

Reviewer #2: (No Response)

4. Have the authors made all data underlying the findings in their manuscript fully available?

Reviewer #1: Yes

Reviewer #2: (No Response)

5. Is the manuscript presented in an intelligible fashion and written in standard English?

Reviewer #1: Yes

Reviewer #2: (No Response)

6. Review Comments to the Author

Reviewer #1: (No Response)

Reviewer #2: (No Response)

7. PLOS authors have the option to publish the peer review history of their article (what does this mean?). If published, this will include your full peer review and any attached files.

Reviewer #1: No

Reviewer #2: No

---

## [Editor Report · Acceptance letter]

12 Mar 2024

PONE-D-23-33530R1 

PLOS ONE

Dear Dr. Mbuthia, 

I'm pleased to inform you that your manuscript has been deemed suitable for publication in PLOS ONE. Congratulations! Your manuscript is now being handed over to our production team.

Kind regards, 

on behalf of

Dr. Patricia Khashayar 

Academic Editor

PLOS ONE